# Genome-Wide Identification and Functional Characterization of *FAR1-RELATED SEQUENCE* (*FRS*) Family Members in Potato (*Solanum tuberosum*)

**DOI:** 10.3390/plants12132575

**Published:** 2023-07-07

**Authors:** Qingshuai Chen, Yang Song, Kui Liu, Chen Su, Ru Yu, Ying Li, Yi Yang, Bailing Zhou, Jihua Wang, Guodong Hu

**Affiliations:** 1Shandong Provincial Key Laboratory of Biophysics, Institute of Biophysics, Dezhou University, Dezhou 253023, China; sog_yag@163.com (Y.S.); dakui1618@163.com (K.L.); 19811807886@163.com (C.S.); angelyuru@163.com (R.Y.); yangyi@dzu.edu.cn (Y.Y.); blzhou@126.com (B.Z.); jhw25336@126.com (J.W.); 2College of Life Science, Dezhou University, Dezhou 253023, China; snly2007@163.com

**Keywords:** potato, genome-wide analysis, StFRS family, expression pattern, abiotic stress

## Abstract

FAR1-RELATED SEQUENCE (FRS) transcription factors are generated by transposases and play vital roles in plant growth and development, light signaling transduction, phytohormone response, and stress resistance. FRSs have been described in various plant species. However, FRS family members and their functions remain poorly understood in vegetative crops such as potato (*Solanum tuberosum*, *St*). In the present study, 20 putative StFRS proteins were identified in potato via genome-wide analysis. They were non-randomly localized to eight chromosomes and phylogenetic analysis classified them into six subgroups along with FRS proteins from Arabidopsis and tomato. Conserved protein motif, protein domain, and gene structure analyses supported the evolutionary relationships among the FRS proteins. Analysis of the *cis*-acting elements in the promoters and the expression profiles of *StFRSs* in various plant tissues and under different stress treatments revealed the spatiotemporal expression patterns and the potential roles of *StFRSs* in phytohormonal and stress responses. *StFRSs* were differentially expressed in the cultivar “Xisen 6”, which is exposed to a variety of stresses. Hence, these genes may be critical in regulating abiotic stress. Elucidating the StFRS functions will lay theoretical and empirical foundations for the molecular breeding of potato varieties with high light use efficiency and stress resistance.

## 1. Introduction

All life on earth directly or indirectly depends on light for growth. Light drives photosynthesis, which provides energy and biomass for the growth of many autotrophic organisms. For plants, light is also an important signaling molecule regulating vegetative and reproductive growth, senescence, metabolism, circadian rhythms, and adaptation to dynamic changes in the environment [1,2]. The importance of transposase-generated transcription factors (TFs) FAR1-RELATED SEQUENCE (FRS) and FRS-RELATED FACTOR (FRF) in light signaling transduction has been recently established and their mechanisms are gradually being elucidated [3].

Under far-red (rather than white) light conditions, the *far-red elongated hypocotyls3* (*fhy3*) mutant is characterized by a long hypocotyl phenotype and was first identified through forward genetics [4]. FHY3 and its homolog FAR-REDIMPAIRED RESPONSE1 (FAR1) are generated by transposases and are typical members of the FRS family [5]. In *Arabidopsis thaliana* (Arabidopsis), the AtFRS family comprises fourteen AtFRSs and four AtFRFs [6]. Most of these contain the *N*-terminal C_2_H_2_ zinc-finger (or the FAR1 DNA-binding) domain, a central putative transposase domain resembling that of MULE transposase, and a *C*-terminal SWIM (SWI2/SNF2 and MuDR transposase) zinc-finger structure [7]. FRS TFs exert their regulatory effects through the FAR1 DNA-binding domain at the *N*-terminal and recognize *cis*-acting (FBS; CACGCGC) elements that induce or repress downstream genes [8,9].

AtFHY3/AtFAR1 directly bind to the promoter of FHY1 or FHY1-LIKE (FHL), activate them, and indirectly participate in the nuclear translocation of the far-red light receptor phytochrome A (phyA) [8,10]. AtFHY3 also directly interacts with the red light receptor phytochrome B (phyB) that affects germination [11]. AtFHY3 cooperates with ELONGATED HYPOCOTYL (HY5) to mediate *CONSTITUTIVE PHOTOMORPHOGENIC1* (*COP1*) induction in response to UV-B light exposure [12]. AtFHY3/AtFAR1 also participate in seedling growth [13], chloroplast division [14,15], chlorophyll biosynthesis [16], starch biosynthesis [17], branching [18], floral development [19,20], circadian rhythm [21,22,23], leaf senescence [24], flowering [23,25], defense response [26], and adaptations to abiotic stress, such as shade avoidance [26,27], phosphate starvation [28], carbon starvation [29], low temperature [30,31,32], osmotic stress [31], and oxidative stress [33]. Additionally, AtFRS7/12 are implicated in glucosinolate biosynthesis and regulate flowering [34,35]. AtFRS4 together with AtFHY3 regulates chloroplast division [14]. However, most of the prior and current research on the functions of FRS family members has focused on light signal transduction.

Plants have evolved conserved mechanisms to maintain a balanced relationship between growing and stressing responses. A number of regulatory factors involved in light signaling have been shown to play important roles in plant stress responses, such as phyA, phyB, HY5 and LIGHT INSENSITIVE PERIOD1 in *Arabidopsis*, G-box BINDING FACTOR1 in wheat and PHYTOCHROME-INTERACTING FACTOR-LIKE14 in rice [36,37,38,39,40,41,42]. The involvement of FRS family members in tea and eucalyptus in responding to stress has been initially reported [30,31]. In contrast, little is known about the mechanisms of action of how FRS is involved in abiotic stress response in other crops.

Potato (*Solanum tuberosum*) is native to the Andes Mountains of South America and is now widely grown globally. It has low average water consumption, high photosynthetic efficiency, strong environmental adaptability, and substantial yield [43,44,45]. Potato is the third largest food crop and the most important tuber crop worldwide. It is a staple food for 1.3 billion individuals and an important source of raw materials for food and industrial processing [46,47,48,49]. The sequencing program conducted on the potato genome provides an asset for genome-wide analysis of the potato FRS family [50]. However, the actions and mechanism of regulation of StFRSs in potatoes have not been scientifically reported.

Here, we investigated the distribution of 20 *StFRS* genes in the potato genome, systematically studied their structures, chromosomal locations, conserved motifs, the *cis*-elements in their promoter regions, their phylogenetic classification, and their functional interaction networks. We also compared the *FRS* homologs in potato, Arabidopsis, tomato, eggplant, pepper, and tobacco. The expression patterns of *StFRS*s in different tissues and under multiple stress conditions were analyzed using transcriptome data. We further investigated the expression of *StFRSs* with different abiotic stresses in the “Xisen 6” cultivar. This work will help elucidate the functions of FRSs and provide the theoretical and practical basis for their use in molecular crop improvement.

## 2. Results

### 2.1. StFRS Identification and Analysis

In this study, a total of 20 *StFRS* genes were identified in the potato genome, and these genes have been named according to the chromosomal loci where they are located (Table 1; Figure 1). The CDS and protein sequences are listed in Appendix A. As shown in Table 1, the StFRS proteins are between 89 aa (StFRS18) and 597 aa (StFRS1) long. The MWs of the StFRSs are between 9.99 kDa (StFRS18) and 66.50 kDa (StFRS1); the predicted pI values for the StFRSs are between 4.38 (StFRS18) and 9.39 (StFRS13); most StFRSs are localized to the cytosols or the nuclei; however, StFRS2 and StFRS12 are localized to the mitochondria while StFRS4 and StFRS17 are localized to the plastids. Hence, the genes at these locations might have undergone novel functional evolution.

### 2.2. Chromosomal Location and Duplication of StFRSs

To understand the distribution of *StFRS* genes in the potato genome, we noted the chromosomal location based on their information from the genome database. As shown in Figure 1, twenty genes were localized on eight chromosomes, including five on chromosome 6, four on chromosome 9, three on chromosome 3, two each on chromosomes 1, 2, and 8, one each on chromosomes 4 and 7, and none on chromosomes 5, 10, 11, or 12.

Tandem and segmental duplications contribute to gene family generation during evolution [51]. There are two clusters of tandem *StFRS* on the potato chromosome, including *StFRS10* and *StFRS11*, and *StFRS12* and *StFRS13* (Figure 1), and these two pairs of gene clusters with tandem duplication belong to the following two subgroups: subgroup II and V, respectively (Figure 3). Segmental duplications are fragments of DNA with nearly identical sequences that are larger than 1 kb [52]. We analyzed synteny blocks among the potato chromosomes to clarify the evolution of potato *StFRS*s through a Multiple Collinearity Scan toolkit X version (MCScanX). Appendix A shows that one pair of segmental duplications (*StFRS9*/*StFRS10*) was identified and their Ka/Ks < 1 (Appendix A). Thus, they might have undergone purifying selection during evolution. Tandem and segmental duplications may play important roles in the evolutionary expansion of *StFRSs*.

### 2.3. Synteny Analysis of FRS in Different Plant Species

To investigate the molecular evolutionary relationships of FRS in plants, we performed synteny analysis between StFRSs and homologs in other species. Figure 2 showed that the numbers of collinear FRS pairs between potato and Arabidopsis, tomato, pepper, eggplant, and tobacco were 8, 11, 5, 8, and 0, respectively. The FRS orthologs syntenic with the StFRSs in other plant species are listed in Appendix A.

The gene pairs most homologous to other species were StFRS9, StFRS10 and StFRS12 on chromosome 6 (Figure 2). Certain StFRSs were syntenic with >2 genes in the Arabidopsis and tomato genomes. Potato StFRS10, StFRS12, and StFRS14 were syntenic with two AtFRSs from Arabidopsis. Solyc03g117520.2.1, Solyc06g068210.2.1, and Solyc06g065940.3.1 in tomato were syntenic with StFRS9. For all direct homolog pairs, Ka/Ks is <1 (Appendix A). Therefore, these genes evolved under the influence of negative or purifying selection. The FRS orthologs in various plants may have facilitated evolution of the *FRS* family. The FRS orthologs between potato and other plants were anchored in conserved syntenic blocks. In addition, no pairs of collinear FRSs were shared between potato and tobacco, suggesting a long-distance phylogenetic relationship between the two species.

### 2.4. Phylogenetic, Gene Structure, Conserved Domain, and Motif Analyses

To investigate the evolutionary relationships of FRSs between potato and the other species, we plotted a maximum likelihood (ML) phylogenetic tree according to the multiple protein sequence alignments of 65 FRSs, including 20 StFRSs from potato, 27 SlFRSs from tomato [53], and 18 AtFRSs from Arabidopsis [6,54]. The phylogenetic distribution indicated that the StFRSs were divided into six groups designated I–VI (Figure 3). Except for subgroup IV, which contained only 1 gene, the other StFRS transcription factor subfamilies contained 3–5 genes in varying numbers (Figure 4A). With the exception of subgroup VI, which consisted of StFRS5, StFRS16, and StFRS20, the subfamily classification of the StFRS proteins closely resembles that of the Arabidopsis and tomato FRS subfamilies, making the results of this classification similar to those of other reported species, including maize, tea, eucalyptus and cotton [30,31,55]. Therefore, investigating the function of StFRS may benefit from looking at other reported species.

We further investigated the StFRS motif distribution, and ten putative conserved motifs among the 20 StFRS members were identified (Figure 4B). Details of these motifs are illustrated in Appendix A. Only the first subgroup comprising StFRS4 and StFRS15 contained all ten motifs, whereas the other subgroups had two to nine motifs. All members of the same phylogenetic group had similar motifs numbers and positions, which was consistent with the numbers and positions of conserved domains (Figure 4C). All StFRSs contained the FHY3 or FAR1 superfamily domains, while the Znf_PMZ domain was identified in FRS2/4/7/9/10/11/15 and the SWIM domain was identified in FRS2/4/7/11/15 (Figure 4C). The protein structures were generally uniform among members of the same subgroup. This observation suggested that they might have similar biological functions. 

We examined the gene intron and exon compositions and exhibited the *StFRS* structures by comparing genomic DNA sequences. Figure 4D shows that three *StFRSs* (*StFRS3*, *StFRS16*, and *StFRS20*) had one exon, six *StFRSs* (*StFRS5*, *StFRS9*, *StFRS11*, *StFRS12*, *StFRS13*, and *StFRS18*) had two exons, and all other *StFRS*s had three to eleven exons. The foregoing results combined with those of the *StFRS* phylogenetic tree analysis (Figure 4A) suggested a similar gene structure within the same subgroup. Hence, the observed differences in gene structure might be related to the biological functions.

### 2.5. In Silico Analysis of Cis-Elements in StFRS Promoters

To provide insight into the regulatory mechanisms, metabolic networks, and functions of the *StFRSs*, the 2 kb upstream sequences of the *StFRS* start codons were used to scan for *cis-*elements. We identified six classes of *cis*-elements involved in plant growth and development of the *StFRS* promoters (Figure 5). These included the GCN4-motif, 3-AF1 binding site, CAT-box, CCGTCC-box, circadian element, and O_2_-site. The promoters of 19/20 *StFRSs* each contained ≥2 phytohormone response elements responding to various signals such as gibberellins, auxins, ethylene, abscisic acid, jasmonic acid, and salicylic acid. Only *StFRS6* lacked any phytohormone response element. Moreover, we identified 16 classes of light response elements among the *StFRS* promoters. In total, 16 of the 20 *StFRSs* contained a total of 43 Box-4 elements and the latter were the most abundant in the present study. Several abiotic and biotic stress response elements such as AREs, MBSs, TC-rich repeats, LTRs, and WUN-motifs were identified among the *StFRS* promoters. Thus, certain *StFRSs* may be implicated in potato responses to drought, hypoxia, low temperatures, and pathogen and predator attack. The preceding findings suggest that *StFRSs* have a wide range of biological functions in potato growth and development in addition to light signaling response.

### 2.6. Protein Interaction Networks of StFRS Proteins

To investigate StFRS potential interactions and characterize their biological functions, the STRING server was used to integrate StFRS into the Arabidopsis association model. As illustrated in Figure 6, the interaction network of StFRS was mapped to the Arabidopsis protein interaction network. There is an existing interaction between the subgroup I of StFRS15 (corresponding to AtFAR1 in Arabidopsis) and StFRS14 (corresponding to AtFHY3), both of which can potentially interact with several other protein types, including phyA, FHY1, FHL and HY5, further demonstrating the important role of StFRS in light signaling. In addition, StFRS proteins can potentially interact with CCA1, ABA3, JAZ3, HDA15 and SPCH among others, suggesting a multifaceted function of StFRS proteins.

### 2.7. StFRS Expression Patterns

The potato transcriptome data from Spud DB were used to analyze *StFRS* expression in different potato tissues. As shown in Figure 7, no reads were detected for *StFRS3*, *StFRS5*, or *StFRS20*. However, the other 17 *StFRSs* were expressed with varying degrees in the roots, stems, leaves, tubers, sepals, carpels, stamens, petals, whole mature flowers, and fruits. *StFRS14* was highly expressed in all tissues tested. By contrast, *StFRS4*, *StFRS7* and *StFRS15* within the same subfamily were expressed at relatively low levels in all tissues and organs. This result indicated that *StFRS14* played relatively more important roles in plant growth and development than the other genes in subgroup I (Figure 7, Appendix A). The specificity of *StFRS* expression in various doubled monoploid (DM) potato tissues and organs correlated with the *cis*-elements in the promoter regulatory region, indicating the wide diversity of *StFRS* functions.

We also analyzed *StFRSs* expression in DM potato subjected to various phytohormones. Figure 8B suggested that abscisic acid (ABA) not only induced most of the *StFRS*s but also strongly modulated *StFRS4* and *StFRS6* expression (Log_2_^FoldChange^ > 1.5). Indole-3-acetic acid (IAA) could induce *StFRS4*/*11*/*13*/*15*, with the strongest ability to induce *StFRS15.* However, it repressed all other *StFRSs*, especially *StFRS10*/*18*/*19* (Figure 8B, Appendix A). Gibberellic acid (GA_3_) inhibited *StFRS4*/*6*/*13*/*14* but promoted other *StFRS*s, especially *StFRS4* with the most significant change (Log_2_^FoldChange^ = −21.27). BAP (6-benzylaminopurine) repressed almost all *StFRS*s, and particularly *StFRS1*/*7*/*13*/*15*. Both ABA and IAA promoted *StFRS4,* whereas GA_3_ inhibited it. Most *StFRSs* and especially *StFRS15* were upregulated in response to salt (150 mM NaCl) and osmotic (260 μM mannitol) stress. By contrast, most *StFRSs* were insensitive to heat treatment. Only *StFRS6* was downregulated, while *StFRS18* was upregulated in response to heat stress (35 °C) (Figure 8C, Appendix A).

We further analyzed the expression patterns of *StFRS*s in response to *Phytophthora infestans* and the elicitors acibenzolar-*S*-methyl (BTH) and *DL*-β-amino-*N*-butyric acid (BABA). BTH mildly repressed most *StFRSs* and BABA was strongly downregulated. *Phytophthora infestans* significantly upregulated *StFRS4,* whereas BABA and BTH downregulated it. BTH induced *StFRS17,* whereas *Phytophthora infestans* and BABA repressed it (Figure 8D, Appendix A).

### 2.8. StFRS Expression Patterns under Abiotic Stresses in “Xisen 6” Potato Cultivar

It has been shown that the light-signaling regulatory proteins play an irreplaceable role in the salt tolerance of plants [40,41,42]. To investigate the role of *FRS* genes in salt tolerance in potato, we selected seven genes with significant changes (Log_2_^FoldChange^ > 1) based on the transcriptomic results of salt-treated DM potatoes (Appendix A), and further analyzed the virus-free seedlings of the “Xisen 6” cultivar. “Xisen 6”, a sexual hybrid selected from “Shepody” and “XS9304”, has shown good yield performance in several parts of China [56], including the saline Yellow River Delta. When treated with different concentrations of NaCl for 12 h, the reverse transcription quantitative polymerase chain reaction (RT-qPCR) assays showed that the transcript level of *StFRS* genes were less affected by 50 mM NaCl, and the transcript levels of the other five genes tested, except *StFRS13* and *StFRS19*, showed differences under different concentrations of salt treatment. For example, both 100 mM and 200 mM NaCl induced an increase in *StFRS15* transcript level, but no significant difference was observed under 150 mM NaCl treatment (Figure 9). These results indicate that *StFRS* genes exhibit differential performance in response to salt stress treatment.

To further investigate the function of *StFRSs* in stress response, transcript levels were analyzed by RT-qPCR under low temperature (4 °C), 20% polyethylene glycol (PEG) and NaHCO_3_ treatment. As illustrated in Figure 10, the 16 *StFRSs* tested showed different responses to these stresses. Low temperature treatment resulted in significant down-regulation of the expression of ten *StFRSs*, while the expression of *StFRS14* and *StFRS15* was up-regulated; PEG treatment resulted in an up-regulation of the expression of *StFRS9*, while the expression of another four *StFRSs* was down-regulated; NaHCO_3_ treatment also resulted in a high expression of *StFRS6*, *StFRS10* and *StFRS13*, while the expression levels of five *StFRSs* were down-regulated. The above results suggest that these family members may have different functions in potato growth and development, as *StFRSs* show different expression responses to the various stresses.

## 3. Discussion

FRS TFs are generated by transposases [7,8]. They have been domesticated and adapted for an extended period of time and play important functions in several other plant biological development processes besides light signaling transduction [3]. Prior research on Arabidopsis FHY3 and FAR1 has clarified the vital roles of FRSs in plant growth and development [54]. There are ongoing studies on the FRSs in several species, such as rice, wheat, tomato, cotton, tea, eucalyptus, silver birch (*Betula pendula*) and *Tibetan Prunus* [30,31,32,53,55,57,58,59]. In the present study, we identified StFRS family members in potato for the first time, identified the biological pathways in which they might participate, and focused on their roles in the abiotic stress response.

### 3.1. Phylogenetic Analysis of the FRS Genes in Potato

We identified twenty *StFRS* genes distributed over eight chromosomes, containing three tandem duplicated gene clusters with a total of seven genes found on chromosomes 6 and 9, which are shared by two subgroups (Figure 1 and Figure 3). The genes derived from the tandem duplications and segmental duplication constituted the StFRS family and might play similar roles in various biological processes [60]. Genome-wide duplication events play a role in maintaining the function of gene families as well as functional expansion, such as the FRS family in other species [6,30,53], and other gene families in potato [61,62,63]. Gene structure and motif composition varied widely between subgroups. Hence, the members of StFRS family are distinct. Thus, further experiments, such as genetic transformation or gene knockout, are required to complete the functional verification of this family.

A phylogenetic analysis revealed that the StFRS subfamily classification of potato resembled that of Arabidopsis and tomato (Figure 3). Thus, the FRS family members have been highly conserved throughout evolution and may have functions similar to those of Arabidopsis or tomato. For these reasons, it may be possible to predict the functions of potato FRS family members based on those determined for Arabidopsis and tomato in prior studies. However, the three genes in the StFRS subfamily VI have undergone substantial evolutionary changes. Analyses of tissue expression patterns through RNA-seq indicated FPKM = 0 for both *StFRS5* and *StFRS20* (Appendix A). We speculate that they may evolve into pseudogenes for undergoing deletion events or functional changes during evolution. 

### 3.2. StFRS Protein Interaction Network

In addition to structural diversity, the StFRS family genes may have other functional diversity. Significantly, *StFRS* promoters contain several types of *cis*-elements, and the functional diversity to which they respond has been demonstrated in other species (Figure 5) [6,30,53]. In addition, StFRS forms functional complexes with other proteins (Figure 6). For example, interactions with HDA15 may contribute to seed germination, hypocotyl elongation and other processes in light signals [64,65,66]. Other potential StFRS–protein interactions exist in potato. However, these require further experimental validation.

### 3.3. StFRS Response to Abiotic Stresses

Although FRS was originally reported to be involved in light signaling, recent studies of the *FRS* gene family have shown that FRS proteins can directly bind to multiple genes (e.g., *FHY1*, *FHL*, *COP1*, *HEMB1*, *EARLY-FLOWERING4*, *CLAVATA3*, *myo-INOSITOL-1-PHOSPHATE SYNTHASE*, *ISOAMYLASE2*, *PHYTOCHROME RAPIDLY REGULATED1* (*PAR1*) and *PAR2*, and *WRKY28*, etc.) in vivo and are involved in regulating multiple physiological processes including phytohormone signaling, biotic/abiotic stress response, flowering and senescence in plants [12,16,17,19,23,24,25,26,27,33,67]. Hormones trigger the regulation of plant growth adaptations in response to drought, salt and other stresses [68]. We found that abiotic and biotic stress can regulate *StFRSs* expression (Figure 8, Figure 9 and Figure 10), but it is unclear that how FRS regulate hormonal signals in response to stresses. One possibility is the activation of the ABA signal pathway, as demonstrated in Arabidopsis [13,69]; other stress-related hormones such as ethylene, salicylic acid, and growth-related hormones such as auxin and gibberellin may also play a role in the regulation of stress via StFRS (Figure 8). The involvement of FRS in plant energy supply is another possibility for its involvement in stress response [29,70], as prolonged abiotic stress inhibits the activity of photosystem II to reduce the energy supply of plant [71].

Recent studies have shown that light is involved in regulating the response of plants to salt stress, and that the translocation of the photoreceptors phyA and phyB into the nucleus is important for salt tolerance in plants [40,41,42], and that the proteins HY5 and PIF4, which interact with FHY3 in Arabidopsis, are also involved in the salt tolerance process [37,40]. Since AtFHY3 is an important factor in allowing the translocation of phytochrome into the nucleus [8], do StFRSs play a role in salt tolerance in potato by affecting the translocation of phytochrome into the nucleus? Alternatively, is it possible that StFRSs regulate salt tolerance via direct binding to the FBS elements of salt response genes? In the present study, we examined *StFRSs* expression patterns in the cultivar potato subjected to salt and alkali stress (NaCl and NaHCO_3_ treatments, respectively) (Figure 9 and Figure 10). SlFHY3 enhances chilling tolerance in tomato [32], and a similar mechanism may be conserved in potato, where StFRS14/StFRS15 cold induction regulate chilling tolerance by integrating environmental light and internal (inositol synthesis) signals. Although preliminary results of the experiments indicate the transcript levels of *StFRS*s are affected by stress, the main question that needs to be addressed in the future is how StFRSs are involved in balancing plant growth and stress resistance.

## 4. Conclusions

We identified 20 putative *StFRSs* in potato by genome-wide analysis and characterized them according to their chromosomal locations, phylogenetic relationships, gene structures, conserved motifs, duplication events, synteny, promoter *cis*-elements, and expression profiles. The StFRSs were divided into five subfamilies that were evolutionarily related to the tomato FRSs. *StFRS* expression profiles revealed that they occur in many potato tissues and are involved not only in light signaling but also in response to environmental stresses, where they may play an irreplaceable role. These findings provided theoretical and empirical bases for improving the agronomic traits of potato and other solanaceous crops such as tomato, eggplant, tobacco, and pepper. Furthermore, with the deciphering of the genome of the highly complex tetraploid potato cultivar [72,73], we will gain a deeper understanding of the mechanisms about StFRSs in potato growth, development and stress resistance, which will provide theoretical guidance for the future development of new potato cultivars with high yield and resistance to climate change.

## 5. Materials and Methods

### 5.1. StFRS Identification in Potato

To identify *StFRS* gene family members in potato (*Solanum tuberosum* L.), the FAR1/FHY3 DNA binding (PF03101), SWIM (PF04434), and MULE transposase (PF10551) domains were extracted from the Pfam database (http://pfam.xfam.org/ (accessed on 15 January 2023)) [74] and used as queries to search among the annotated potato protein sequences (DM v. 6.1) in Spud DB Potato Genomics Resources (http://spuddb.uga.edu/ (accessed on 15 January 2023)) [50]. To this end, the Hmm search program in HMMER software (https://www.ebi.ac.uk/Tools/hmmer/search/hmmsearch (accessed on 15 January 2023)) was used at the cutoff = 0.01 [75]. For multiple spliced transcripts, the longest was selected for analysis. The StFRS protein sequences were verified in the NCBI Conserved Domains Database (CDD; http://www.ncbi.nlm.nih.gov/Structure/cdd/wrpsb.cgi (accessed on 19 January 2023)) [76] and on the SMART website (http://smart.embl-heidelberg.de (accessed on 19 January 2023)) [77]. Proteins lacking the FRS conserved domain were removed.

### 5.2. StFRS Sequence Analysis and Characterization

Expasy (http://www.expasy.org/ (accessed on 19 January 2022)) was used to quantify the amino acid numbers, molecular weights (MW), and theoretical isoelectric points (pI) of StFRS proteins [78]. The subcellular localizations of the latter were predicted using WoLF PSORT II (http://www.genscript.com/wolf-psort.html (accessed on 19 January 2023)) [79].

The conserved motifs of the StFRS proteins were analyzed using the Simple MEME Wrapper in TBtools, with a maximum of 10 misfits and an optimal motif width of 6–50 amino acid residues, with any number of repetitions [80]. The *StFRS* structures and chromosomal locations were analyzed and displayed using Gene Structure View and Gene Location Visualize in TBtools, respectively [80].

### 5.3. Prediction of Cis-Elements and Protein Interaction Networks

The sequence 2 kb upstream of the *StFRSs* was extracted from Spud DB. The PlantCARE website (http://bioinformatics.psb.ugent.be/webtools/plantcare/html/ (accessed on 19 January 2023)) was consulted to screen potential *cis*-elements in the *StFRS* promoters [81]. The functional interaction networks of StFRS family proteins were predicted by STRING (https://string-db.org/ (accessed on 16 June 2023)), using the protein sequences of the StFRSs to search the potato and Arabidopsis databases.

### 5.4. Phylogenetic Tree Construction

The Arabidopsis AtFRS protein sequences were extracted from The Arabidopsis Information Resource (TAIR) (https://www.arabidopsis.org/ (accessed on 23 January 2023)). The tomato (*Solanum lycopersicum*) SIFRS protein sequences were extracted from the Sol Genomics Network (SGN) (https://solgenomics.net/ (accessed on 23 January 2023)). Evolutionary analyses were conducted with the One Step Build a ML Tree in TBtools [80]. Unrooted trees were constructed using the maximum likelihood (ML) method with 1000 bootstrap replicates [82]. The online software program iTOL (https://itol.embl.de/ (accessed on 23 January 2023)) was used to embellish the phylogenetic tree [83].

### 5.5. Gene Duplication and Synteny Analysis

The potato, Arabidopsis, and tomato genome sequences were obtained through the database mentioned above. The genomes of pepper (*Capsicum annuum*) and tobacco (*Nicotiana tabacum*) were obtained from EnsemblPlants (http://plants.ensembl.org/info/data/ftp/index.html (accessed on 23 January 2023)) [84]. *StFRS* gene duplications were identified using the following criteria: (1) shorter gene sequence length coverage > 70% of the longer sequence; and (2) >70% similarity of both sequences when compared [85,86]. Two genes from the same family separated by five or fewer genes in a 100 kb chromosomal fragment are considered tandem duplicates [87]. Gene duplication and synteny analysis were performed using MCScanX (https://github.com/wyp1125/MCScanX (accessed on 23 January 2023)) [88] and visualized with TBtools [80]. Synonymous (Ks) and non-synonymous (Ka) *StFRS* duplication events were calculated using KaKs_Calculator v. 2.0 (https://sourceforge.net/projects/kakscalculator2/ (accessed on 23 January 2023)) [89].

### 5.6. Transcriptome Analysis

The publicly available transcriptome dataset was obtained from Spud DB to analyze the expression patterns of 20 *StFRSs* in various tissues of doubled monoploid (DM) potato [50,90]. The *StFRS* expression patterns were analyzed in whole potato plants subjected for 24 h to various abiotic stress such as 150 mM NaCl (salt), 260 μM mannitol (drought), 35 °C (heat), and different phytohormones including 50 μM ABA (abscisic acid), 10 μM IAA (auxin), and 50 μM GA_3_ (gibberellic acid). The biotic stress treatments comprised *Phytophthora infestans* (*P. infestans*) infection and the elicitors acibenzolar-*S*-methyl (BTH), and *DL*-β-amino-*N*-butyric acid (BABA) for 24 h, 48 h, and 72 h, respectively. Accession numbers of transcriptomic data from Spud DB in the present study are listed in Appendix A.

### 5.7. Plant Materials and Growth Conditions

Freshly cut single-node potato plants (*Solanum tuberosum* “Xisen 6” cultivar) were placed into a culture bottle containing ½ Murashige and Skoog (MS) medium with 1% (*w/v*) sucrose at 25 °C under a 16 h light/8 h dark photoperiod. Plant material treatments from previously reported studies with slight modifications were used [62,91]. For the salt, alkali and drought treatments, the 14-day-old seedlings roots were exposed to the liquid medium containing 0 mM, 50 mM, 100 mM, 150 mM or 200 mM NaCl, 50 mM NaHCO_3_, and 20% (*w*/*v*) PEG 6000, respectively. For the cold treatment, the 14-day-old seedlings grown under normal conditions were transferred to an artificial climate chamber maintained at 4 ℃. All seedlings were harvested at 12 h by liquid nitrogen, with three biological replicates per treatment, and stored at −80 °C until further analysis. 

### 5.8. Total RNA Isolation and Expression Analysis

Whole potato seedlings subjected to the NaCl treatment were ground in liquid nitrogen and their total RNA was isolated with a Biospin Plant Total RNA Extraction Kit (BSC65S1B, Hangzhou Bioer Technology, Hangzhou, China). Then, 1 μg of RNA per sample was reverse-transcribed into cDNA with an All-in-One First-Strand Synthesis MasterMix Kit (EG15133S; Yugong Biolabs, Lianyungang, China). The real-time fluorescence quantitative PCR (RT-qPCR) protocol used was described in previous works [92]. All primers used in the present study are listed in Appendix A.

### 5.9. Statistical Analysis

All samples were analyzed in triplicate. Data are presented as mean ± standard deviation (SD), unless otherwise specified. Statistical analyses were performed using Excel 2019 (Microsoft Corp., Redmond, Washington, USA). Statistically significant differences between treatment means were determined by Student’s *t-*test. ns was considered no significant, * *p* < 0.05 was considered significant and ** *p* < 0.001 was considered extremely significant.

## Figures and Tables

**Figure 1 plants-12-02575-f001:**
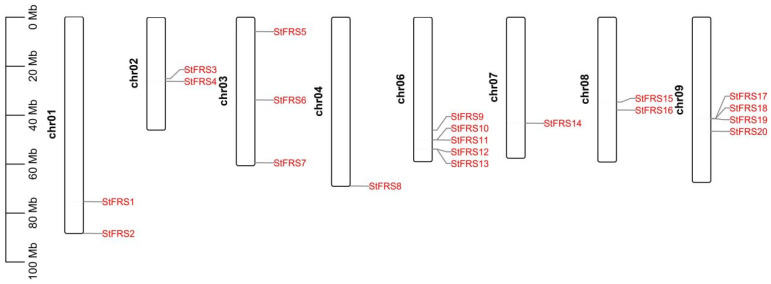
Genomic distribution of *StFRSs* on potato chromosomes. Chromosome numbers are on the left and *StFRSs* are on the right of chromosomes. Scale bar on the left indicates chromosome length.

**Figure 2 plants-12-02575-f002:**
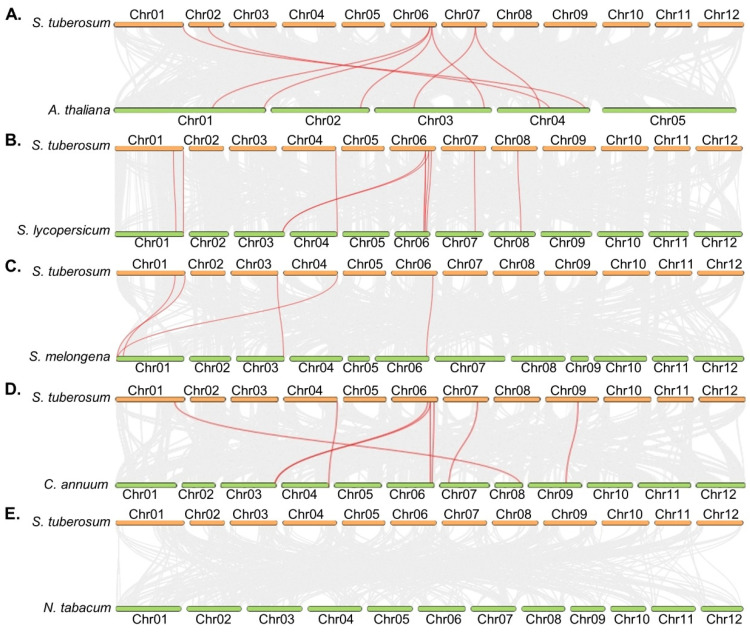
Synteny analysis of *StFRSs* between potato and five other plant species. Gray lines at bottom indicate collinear blocks with potato and other plant genomes. Red lines indicate *FRS* pairs. Results of synteny analyses between potato and Arabidopsis (**A**), tomato (**B**), eggplant (**C**), pepper (**D**), and tobacco (**E**).

**Figure 3 plants-12-02575-f003:**
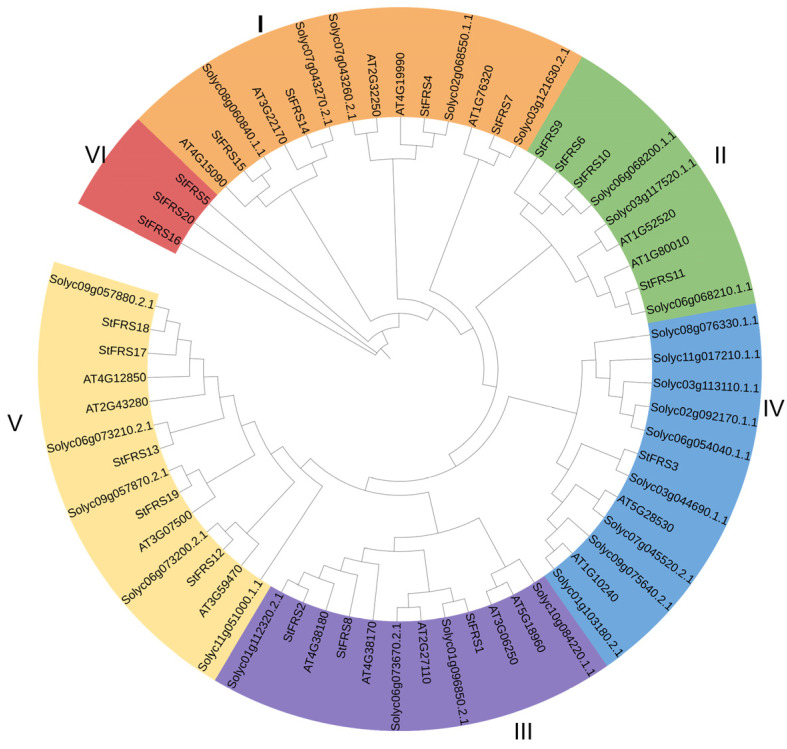
Phylogenetic analyses of StFRS proteins. Phylogenetic relationships among 65 FRS proteins in potato, Arabidopsis, and tomato. Tree divided FRS proteins into subgroups. Phylogenetic tree constructed in TBtools using ML method and based on complete FRS protein aa sequences. Bootstrap = 1000.

**Figure 4 plants-12-02575-f004:**
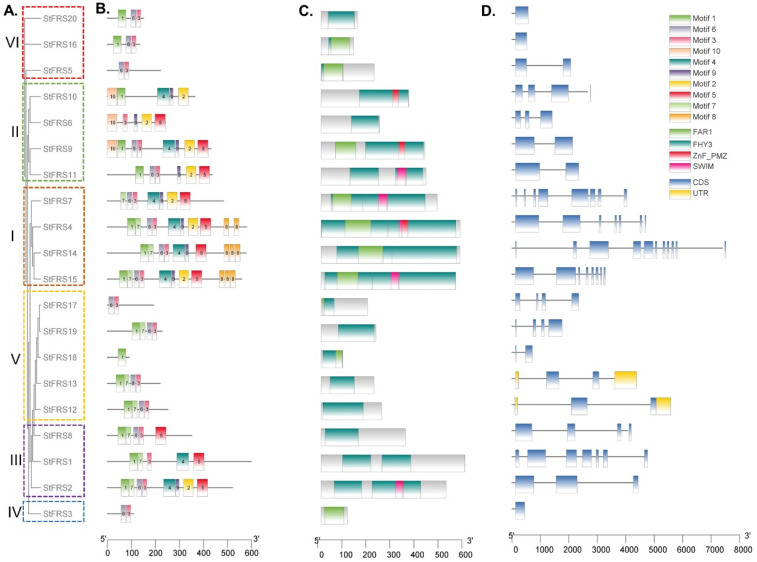
Phylogenetic, comparative motif, conserved domain, and structural analyses of StFRSs. (**A**) Phylogenetic tree constructed using StFRS aa sequences. Orange: subgroup I. Green: subgroup II. Purple: subgroup III. Blue: subgroup IV. Yellow: subgroup V. Red: subgroup VI. (**B**) StFRS motif analysis. Top 10 motifs identified from potato protein obtained by MEME analysis through TBtools. (**C**) Conserved domain analysis of StFRS proteins. Green, lake blue, pink, and yellow rectangles represent FAR1, FHY3, Znf_PMZ, and SWIM domains, respectively. (**D**) Structural analysis of *StFRS* genes. Yellow rectangles, blue rectangles, and black lines represent UTR, CDS, and introns, respectively. Scale bar at bottom estimates lengths of protein (**B**,**C**) and gene (**D**) sequences.

**Figure 5 plants-12-02575-f005:**
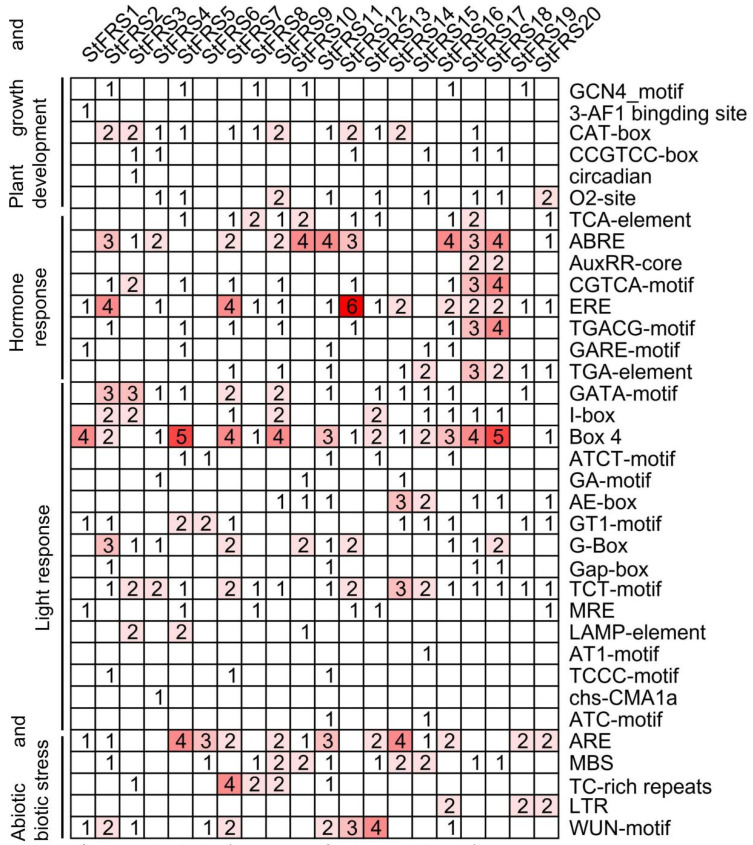
*Cis*-acting elements predicted in *StFRS* promoters. *StFRS* promoter sequences (2 kb) were used to predict *cis*-acting elements on PlantCare server. Thirty-five *cis*-acting elements were categorized by function into four clusters.

**Figure 6 plants-12-02575-f006:**
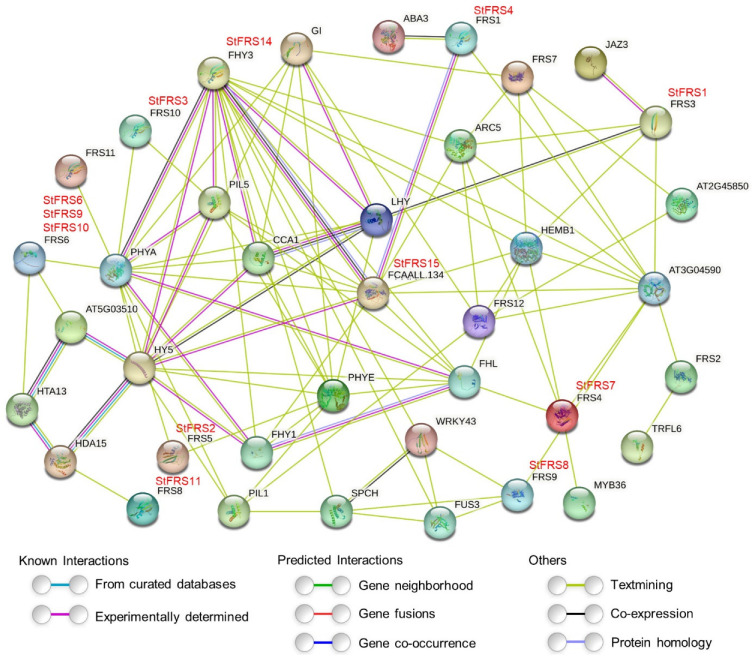
Putative protein interaction network of StFRSs. The homologous proteins in potato and Arabidopsis are shown in red and black, respectively. The colors of the lines indicate the different types of evidence for the prediction of the protein interactions network.

**Figure 7 plants-12-02575-f007:**
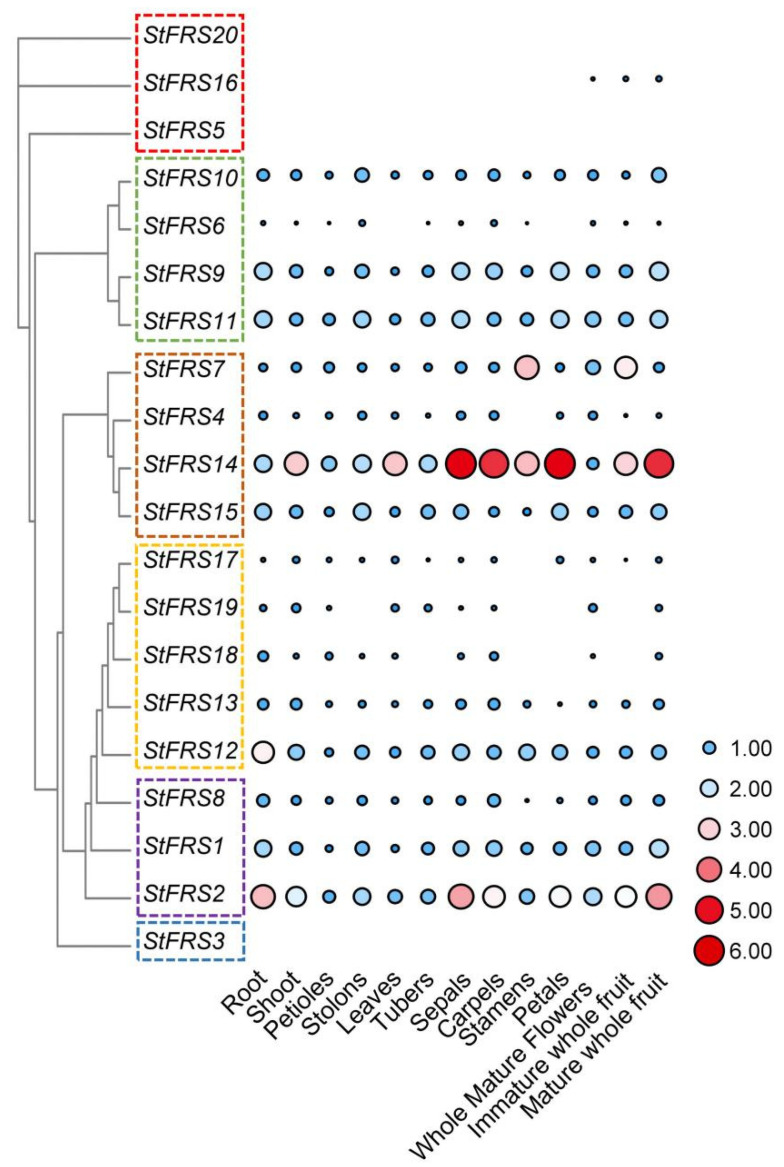
*StFR*S expression patterns in various potato tissues. Thirteen developmental tissues were used to analyze expression patterns. These included roots and shoot of in vitro-grown plants, vegetative organs (petioles, stolons, leaves, and tubers) and reproductive organs (sepals, carpels, stamens, petals, whole mature flowers, and immature and mature whole fruit) derived from greenhouse-grown plants. Color scale represents Log_2_^FPKM^ of *StFRSs*. Blank: no reads; red: high expression level; blue: low expression level. Initial *StFRS* expression levels are listed in Appendix A.

**Figure 8 plants-12-02575-f008:**
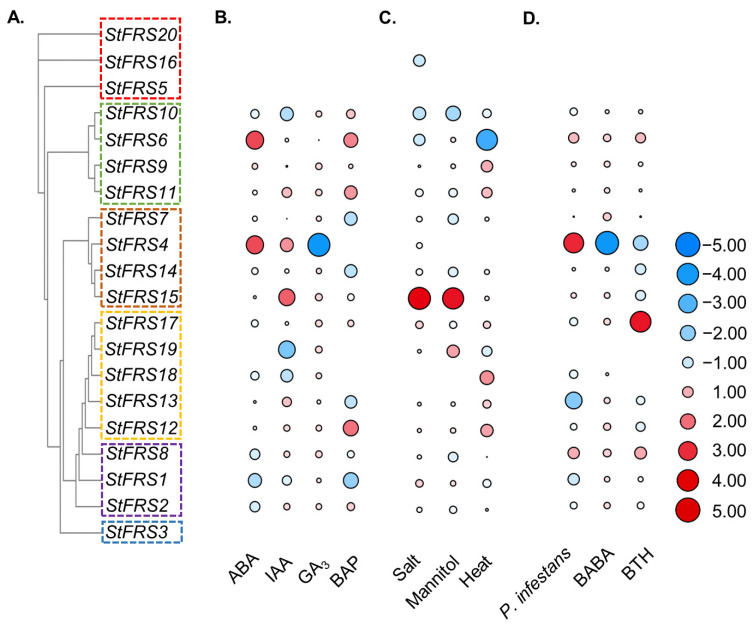
*StFR*S expression patterns in response to different treatment. (**A**) Phylogenetic tree of StFRS. *StFRS* expression patterns in whole potato plants after treated with phytohormones (**B**), abiotic stress (**C**), and biotic stress (**D**). Color scale represents normalized Log_2_^FoldChange^ data.

**Figure 9 plants-12-02575-f009:**
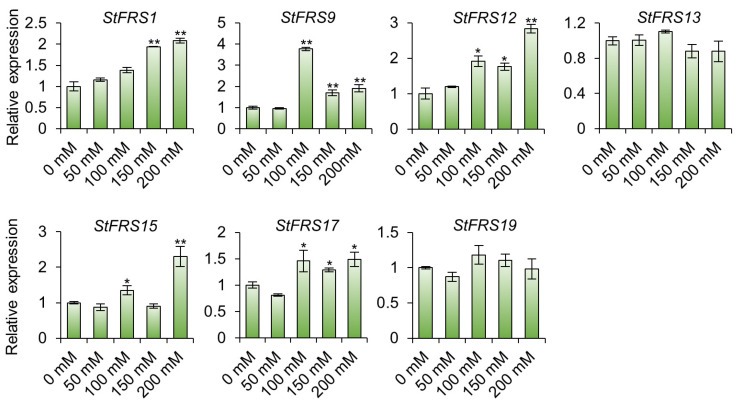
The expression pattern of *StFR*S genes under different concentrations of salt treatment. RT-qPCR assays showing expressions of *StFRS* genes in Xisen 6 after 12 h of treatment with different concentrations of NaCl, with seedlings treated with 0 mM NaCl as control. Mean ± SD (n = 3). * *p* < 0.05, ** *p* < 0.001.

**Figure 10 plants-12-02575-f010:**
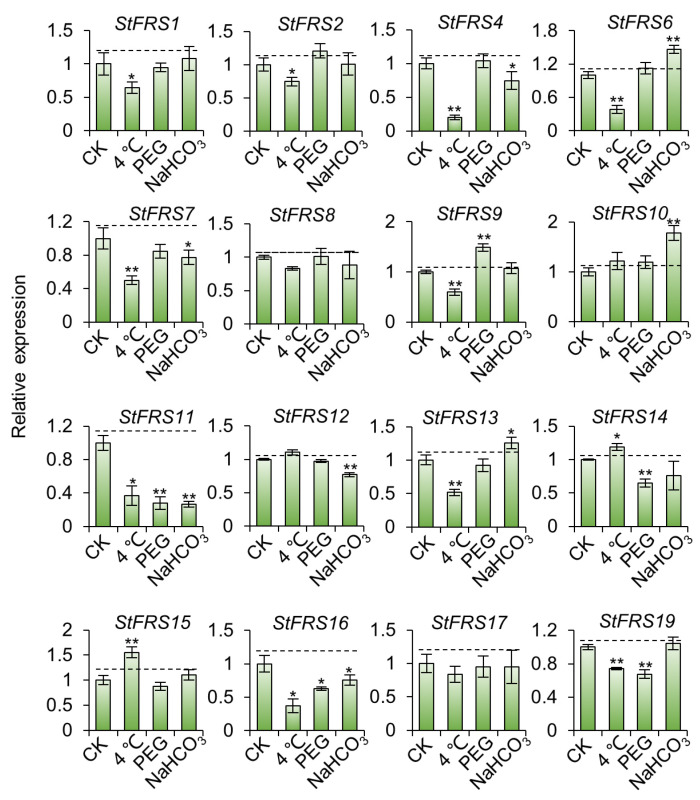
The expression pattern of *StFRS* genes under abiotic stresses. RT-qPCR assays showing expressions of *StFRS* genes in Xisen 6 after 12 h of treatment with different treatments (low temperature, PEG, and NaHCO_3_), and seedlings with no treatment as control. Mean ± SD (n = 3). * *p* < 0.05, ** *p* < 0.001.

**Table 1 plants-12-02575-t001:** *StFRS* family members identified in potato genome.

Gene Name	Locus ID	Chromosome Location	Protein Length (aa)	Protein MW (Kda)	pI	Subcellular Localization
*StFRS1*	Soltu.DM.01G036020	chr01	597	66.5	6.19	cytosol
*StFRS2*	Soltu.DM.01G052050	chr01	520	58.66	5.89	mitochondrion
*StFRS3*	Soltu.DM.02G010140	chr02	108	12.67	8.99	nucleus
*StFRS4*	Soltu.DM.02G011380	chr02	578	64.32	7.86	plastid
*StFRS5*	Soltu.DM.03G005140	chr03	220	25.21	9.41	nucleus
*StFRS6*	Soltu.DM.03G012450	chr03	242	28.13	4.96	cytosol
*StFRS7*	Soltu.DM.03G036430	chr03	482	54.71	8.57	cytosol
*StFRS8*	Soltu.DM.04G038110	chr04	350	39.39	5.86	nucleus
*StFRS9*	Soltu.DM.06G019480	chr06	430	49.89	5.8	nucleus
*StFRS10*	Soltu.DM.06G023880	chr06	363	41.85	4.93	nucleus
*StFRS11*	Soltu.DM.06G023890	chr06	435	49.04	5.58	nucleus
*StFRS12*	Soltu.DM.06G028550	chr06	251	28.91	5.98	mitochondrion
*StFRS13*	Soltu.DM.06G028560	chr06	219	25.1	9.39	cytosol
*StFRS14*	Soltu.DM.07G014070	chr07	576	65.54	5.86	nucleus
*StFRS15*	Soltu.DM.08G011470	chr08	559	63.09	6.35	nucleus
*StFRS16*	Soltu.DM.08G012540	chr08	134	15.28	9.59	nucleus
*StFRS17*	Soltu.DM.09G015460	chr09	192	20.7	6.18	plastid
*StFRS18*	Soltu.DM.09G015470	chr09	89	9.99	4.38	cytosol
*StFRS19*	Soltu.DM.09G015500	chr09	227	25.49	8.13	nucleus
*StFRS20*	Soltu.DM.09G016480	chr09	150	17.02	6.9	cytosol

## Data Availability

Data from this study are available in the article and Appendix A.

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
