# Peer review of "Genome-Wide Identification and Functional Characterization of FAR1-RELATED SEQUENCE (FRS) Family Members in Potato (Solanum tuberosum)"

_plants, 2023, doi:10.3390/plants12132575_

Round 1

Reviewer 1 Report

In this article authors have tried to identify genes in FRS family in Solanum tuberosum. They have identified 20 StFRS genes throughout the genome and did bioinformatics-based in-silico analysis. This in-silico based genome-wide analysis of FRS gene family is overall a fine effort by the authors. In the introduction authors address the question that is most research on FRS family focuses on light signal transduction and little is known about its roles in salt stress response. But only a small portion of this article is about expression changes of StFRS under salt stress. One other major issue of this article is the discussion almost solely focuses on its own data. It lacks connection to what already known about FRS genes in other species and their functions. Below are other comments.

1.      Section 2.2 Chromosomal location and duplication of StFRSs. Explanation is needed for why genes in the three clusters (lines 109-111) are from tandem duplication. And why StFRS9 and StFRS11 are segment duplicates.

2.      Section 2.4 Phylogenetic, gene structure, conserved domain, and motif analyses. The tree only contains three species. I do not know how this conclusion “Overall, the StFRSs strongly resembled the FRSs of Arabidopsis and tomato” (lines 147-148) is made. “The results of the classification were consistent with those of the phylogenetic trees plotted for other plant species” (lines 150-151), where are the other trees?

3.      Table 1. Is StFRS18 a functional protein? Why is the size of StFRSs so different?

4.      Why do authors select Xisen 6, which exhibits mild salt tolerance?

5.      Supplemental figures need descriptive legends.

Author Response

These comments are valuable and helpful in improving our manuscripts, as well as guiding our research. For the revisions, please see the attachment.

Reviewer 2 Report

Dear Authors,

Thank you for submitting the MS. Here are some points I suggested to increase the worth of this paper. Genome wide analysis is quite routinely done for any gene and plant. We need to give a newness to it.

Rephrase the importance of this study "However, FRS family members and their functions remain to be identified and analyzed in potato (Solanum tuberosum, St)". Is this highlighting the importance really? Hihglight in line 71..litte is known about its role in vegetative crops etc etc.

Line 123: Its Synthetic or syntenic? Why was this study carried out just to show closeness with tomatoes? Make this part of the section stronger.

Why was 100mM taken for checking time points for Fig 8 and in the next figure 9 different conc of NaCl. For potato as per literature 150 mM is enough salt stress. I would suggest putting Fig 8 in supplementary and just keep Fig 9 in main text.

Can any hormone or other stress treatment provide to these seedlings? As in ABA? This paper is just salt characterization and cannot be justified as functional characterization.

Can a protein interaction model be generated/propsed to show the importance of the selective salt regulated FRS gene? A model for mechanism of stress tolerance.

For RT PCR only the root tissues would have been sufficient? Justify why the entire seedling was taken for RNA isolation.

Author Response

Those comments are all valuable and very helpful for improving our manuscript, as well as guiding our research. For the revisions, please see the attachment.

Round 2

Reviewer 1 Report

Thanks to authors for making changes according to reviewers’ comments in such short period of time. In the revised manuscript, authors add protein-protein interaction network analysis and investigate expression patterns of StFRS under low temperature, osmotic stress, and alkaline stress. I suggest that authors look deeper into their protein-protein interaction data, transcriptomic analysis, cis-element analysis, and RT-qPCR data and discuss whether one analysis supports the others. Other comments include: 1) Lines 115-117. The definition of tandem duplication is still not clear. 2) Line 131. In the revised version, proteins are used to perform syntenic analysis, instead of DNA sequences mentioned in the old version. Make sure what is written in the revision is not a mistake. 3) Please check thoroughly to correct gene names that are not italicized and protein names that are italicized. There are many of them. 4) Methods still lack details. If default settings are used, please mention. 

Author Response

(The authors gave the same response as above.)

Reviewer 2 Report

Dear Authors,

Thank you for revising the manuscript. All the queries have been addressed well. It is a good read now.

Author Response

Thank you very much for your comments!

Round 3

Reviewer 1 Report

2.3 Synteny analysis of FRS in different plant species. Correct gene names to italic format.